# Contribution of Extracellular Vesicles and Molecular Chaperones in Age-Related Neurodegenerative Disorders of the CNS

**DOI:** 10.3390/ijms24020927

**Published:** 2023-01-04

**Authors:** Leila Noori, Kamila Filip, Zohreh Nazmara, Simin Mahakizadeh, Gholamreza Hassanzadeh, Celeste Caruso Bavisotto, Fabio Bucchieri, Antonella Marino Gammazza, Francesco Cappello, Maciej Wnuk, Federica Scalia

**Affiliations:** 1Department of Biomedicine, Neuroscience and Advanced Diagnostics (BIND), University of Palermo, 90127 Palermo, Italy; 2Department of Anatomy, School of Medicine, Tehran University of Medical Sciences, Tehran 1417653761, Iran; 3Department of Biology, Institute of Biology and Biotechnology, College of Natural Sciences, University of Rzeszow, 35959 Rzeszow, Poland; 4Department of Neuroscience and Addiction Studies, School of Advanced Technologies in Medicine, Tehran University of Medical Sciences, Tehran 1417653761, Iran; 5Department of Anatomy, School of Medicine, Alborz University of Medical Sciences, Karaj 3149779453, Iran; 6Euro-Mediterranean Institute of Science and Technology (IEMEST), 90139 Palermo, Italy; 7Department of Biotechnology, Institute of Biology and Biotechnology, College of Natural Sciences, University of Rzeszow, 35959 Rzeszow, Poland

**Keywords:** central nervous system, neurodegeneration, aging, chaperones system, extracellular vesicles

## Abstract

Many neurodegenerative disorders are characterized by the abnormal aggregation of misfolded proteins that form amyloid deposits which possess prion-like behavior such as self-replication, intercellular transmission, and consequent induction of native forms of the same protein in surrounding cells. The distribution of the accumulated proteins and their correlated toxicity seem to be involved in the progression of nervous system degeneration. Molecular chaperones are known to maintain proteostasis, contribute to protein refolding to protect their function, and eliminate fatally misfolded proteins, prohibiting harmful effects. However, chaperone network efficiency declines during aging, prompting the onset and the development of neurological disorders. Extracellular vesicles (EVs) are tiny membranous structures produced by a wide range of cells under physiological and pathological conditions, suggesting their significant role in fundamental processes particularly in cellular communication. They modulate the behavior of nearby and distant cells through their biological cargo. In the pathological context, EVs transport disease-causing entities, including prions, α-syn, and tau, helping to spread damage to non-affected areas and accelerating the progression of neurodegeneration. However, EVs are considered effective for delivering therapeutic factors to the nervous system, since they are capable of crossing the blood–brain barrier (BBB) and are involved in the transportation of a variety of cellular entities. Here, we review the neurodegeneration process caused mainly by the inefficiency of chaperone systems as well as EV performance in neuropathies, their potential as diagnostic biomarkers and a promising EV-based therapeutic approach.

## 1. Introduction

A considerable number of people develop central nervous system (CNS) diseases, but there is still no appropriate treatment for them [1]. In fact, despite many studies having investigated the pathogenic mechanisms of these diseases over the past years, the lack of sufficient and efficient biomarkers precludes an effective early diagnosis, and therefore, in most cases, the prognosis is poor [2]. Various etiologic factors are involved in the development of neurologic disorders and many of them are known as the consequence of multi-etiopathogenic processes. A certain number of neurological disorders, including nervous system tumors, can be caused by inheritable mutations or are the result of non-inherited sporadic random gene mutations [3,4,5,6,7]. The role of risk factors such as old age, oxidative stress, inflammation, immune and metabolic diseases, and genetic polymorphisms have been indicated in the progression of non-inherited neurological diseases and hold a dominant role in tumor outcome [8]. Alleviated genetic repair potency during ageing and pro-oncogenic tissue alternations induced by senescence collectively lead to an abundance of mutations, and in the meantime, enhance the chance of cancer in old age [9]. Neuroinflammation, a distinctive feature of numerous neurodegenerative diseases, is caused by microglial activation in the CNS [10]. The excessive reaction of the microglial cells to different stimuli, including injury, autoimmunity, infections, chemical substances and neurodegeneration of nervous tissue, leads to overexpression of the immune response, establishing the chronic pathogenic condition observed in neurodegenerative disorders [11]. Recently, neuroinflammation has been described as a double-edged sword that imposes both adverse and constructive effects on the neurons. Although there are many indications of the neurotoxic nature of microglia, others suggest that neuroinflammation is certainly helpful in established conditions to eliminate toxic aggregated proteins and cell debris from the CNS, and triggers the remyelination and secretion of neurotrophic factors [12,13]. In vitro and in vivo studies supported the idea that the accumulation of misfolded and aggregated proteins in the brain causes a series of actions that contribute to neuronal degeneration [14,15]. The appearance of misfolded and aggregated forms of specific proteins in certain neuroanatomical regions is a typical characteristic of neurodegenerative diseases [16].

Molecular chaperones have been shown to be crucial modulators of aggregation and toxic conformation of proteins during neurodegenerative diseases [17]. They particularly detect non-native proteins and assist their proper refolding or facilitate their degradation (Figure 1). 

Notably, an age-dependent growing deficiency in protein quality control (PQC) machinery leads to a reduction in the ability of molecular chaperones to recognize and eliminate potentially toxic aggregations, accumulation of misfolded proteins and disturbed proteostasis, which imply a pathogenic impact on the nervous system and contribute to neurodegenerative disorders [18]. The migration of the misfolded proteins from one site to another during the development of the diseases indicates that their relocation plays an important role in pathological process [19]. The development of knowledge illustrates that extracellular vesicles (EVs) are the chief mediator of intercellular interactions in the nervous system [20,21]. Recent reports propose the association of EVs with various age-induced neurodegenerative diseases, as these nanoparticles have been released by all human cells, including cells in the CNS [22]. The prominent role of EVs in different forms of cell communication, such as neuron-to neuron, neuron-to-glia, glia-to-glia, and glia-to-neuron within the CNS [23], suggest their critical functions in pathogenesis as being potent carriers of aggregated toxic proteins, inflammatory factors and other disease-causing agents among cells [24,25,26]. While the knowledge of the role of EVs in ageing and age-related disease, specially neurodegeneration, is still in its beginning, interest in these fields are growing rapidly [27]. In this review, keeping an eye on the role of chaperone systems (CS) in proteostasis, we explore the present knowledge of the contribution of EVs in the CNS age-related diseases (Figure 2). In addition, a brief statement of the therapeutic potential of EVs is also included.

## 2. The Chaperone System (CS)

Chaperones are highly conserved proteins from bacteria to mammals. Cells rely on a subset of constitutively expressed chaperones that contribute to the ongoing quality control of proteins either found in the cytosol or in intracellular compartments [28]. The CS consists of various molecules such as heat shock proteins (Hsps), which are expressed in stress conditions (e.g., high temperatures, oxidative stress, chemical stress, etc.), chaperones, cochaperones, chaperone cofactors, interactors and receptors. Although most of them exist as oligomers, molecular chaperone families are categorized according to the molecular mass of their monomers. Given that, Hsps molecular chaperones are classified according to their molecular weight (Table 1). 

The Hsp10, Hsp60, Hsp70, Hsp90, Hsp100, and small Hsps (sHsps) families are the most significant ones. Every family typically has multiple subfamilies, which are categorized by cellular localization, function, and/or expression pattern. Members of each family can be found in various cellular compartments, and they may act to protect or regulate a particular group of proteins. Several kinds of chaperones can identify some prevalent aminoacidic pattern in the proteins, such as hydrophobic site or posttranslational modifications, while other chaperones are quite specialized in monitoring a very small set of proteins in which they detect a specific binding site, for example, those cooperating with collagen or regulating actin and intermediate filaments [29]. Indeed, co-chaperones are more prevalent than chaperones. While there are no appreciable differences in the amounts of chaperones between simple and complex species, as an organism becomes more complex, the quantity and variety of co-chaperones increases substantially. This outstanding quality might be required for the quality control system to handle the intricate intracellular structure of superior organisms [30]. Distinct functions for classic molecular chaperones have been identified, including binding sites to the client protein, interaction sites with other chaperones to create a chaperoning team, and interaction sites with subunits of other chaperoning teams to form a chaperoning network. All of these sites are considered fundamental for chaperone efficacy. Moreover, ATP-binding domains in many chaperones provide a source of energy through ATP hydrolysis [31]. A class of molecular chaperones are called chaperonins, which are large double-ring complexes of 800–1000 kDa and consist of 7–9 subunits per ring. The protein folding performed by chaperonins occurs in an ATP-dependent manner in the enclosed central cavity of the double-ring complex [32,33]. Chaperonins are classified as group I, known as GroEL/GroES in prokaryotes, Hsp60/Hsp10 in mitochondria and Rubisco subunit binding protein in chloroplasts, and group II, known as Mm-cpn and thermosome in archaea and the chaperonin-containing TCP-1, also called CCT complex (TRiC), in eukaryotes [3,34]. CCT is a cytosolic chaperonin which, in collaboration with Hsp70 and its cochaperone, the prefoldin protein, contributes to the folding process of about 15% of cytosolic proteins, considering numerous essential structural and regulatory ones [35,36]. In particular, numerous pathological conditions, including those of the nervous system, have been linked to impairments in chaperone function [37]. Deficient chaperones known as chaperonopathies can contribute to pathogenesis in a number of different qualitative and quantitative ways, including that where the chaperone is structurally or functionally defective itself (chaperonopathy by defect) or gene dysregulation results in abnormal increase in chaperone synthesis (chaperonopathy by excess) which can be genetic or acquired. Moreover, in some cases, despite being normal chaperones, they may assist cells that advance pathogenesis; this is considered chaperonopathy by mistake [38]. Imbalanced chaperones are indicative of various cellular efforts to adapt to the pathogenic environment and maintain cellular homeostasis, even in the early stages of pathogenesis [28]. 

## 3. Extracellular Vesicles (EVs)

With the words “Extracellular Vesicles” (EVs), researchers usually refer to various subtypes of cell-released, membranous structures known as exosomes, microvesicles (MVs), microparticles, ectosomes, oncosomes, apoptotic bodies, and by many other names (Figure 3). 

It is referring to their size (prefix micro or nano: microparticles, microvesicles (MVs), nanovesicles, nanoparticles), their cell or tissue of origin (prostasomes, oncosomes), their proposed functions (calcifying matrix vesicles, argosomes, tolerosomes), or simply their presence outside the cells (prefix exo- or ecto-: ectosomes, exosomes, exovesicles, exosome-likevesicles) [39]. Exosomes are considered as endosomal origin particles while plasma membrane-derived particles are called shedding vesicles or ectosomes, including microvesicles or microparticles (Table 2).

Furthermore, EVs are classified according to their size as small <100 nm or <200 nm and medium/large >200 nm. Biochemical composition (e.g., CD63+/CD81+-EVs), emerging condition or cell of origin (podocyte EVs, hypoxic EVs, large oncosomes, apoptotic bodies) are also contributors in their nomenclature [58]. Among them, MVs and exosomes are two dominant subtypes which have been profoundly investigated and classified according to their respective sizes, shapes, biogenesis, origins, and composition. MVs are described as particles of about 100–1000 nm in diameter and directly generated from the cell membrane while exosomes with 30–100 nm in size are formed as intraluminal vesicles and arise from MVBs [59]. Exosomes are present in a considerable number of physiological fluids, including blood, saliva, breast milk, cerebrospinal fluid (CSF), amniotic fluid, urine and seminal fluid and also in pathological samples such as malignant effusions, ascitic and bronchoalveolar fluids, indicating their pivotal involvement in cell talk [60,61]. Extracellular vesicles store and carry bioactive cargo such as proteins, lipids, and different types of nucleic acids including coding and noncoding RNAs between all types of cells (Figure 4) [62]. 

These tiny bilayer membrane particles are actively generated by all human cells and mediate cell-to-cellcommunication during physiological and pathological conditions [63,64].

While at first thought as disposal system of the cells to remove unwanted structures such as misfolded proteins, further studies established their functional role in cellular homeostasis [49]. EVs are considered diagnostic biomarkers reflecting their original cell state; moreover, they are basic agents in various and complicated matrixes of intercellular communication [65]. Depending on the source secreting cell type, their main actions on the target cells and their cargo composition may vary, including cytokines [66], hormones [67], transcription factors, growth factors [68] and heat shock proteins [14]. The released EVs affect target cells via transferring the bioactive molecules to neighboring cells or even toward distal organs [62]. EVs facilitate the passage of cargoes such as membrane proteins and several types of RNA that are normally inhibited by plasma membranes. This potential property of EVs in the intercellular transport of such molecules through the nervous system leads to a novel understanding of intercellular communication in the brain [69].

## 4. EVs in/of CNS

EVs can be released by most cell types, including neurons and glial cells, and participate in cell communication and signaling pathways under healthy and pathological conditions of the CNS. In the CNS, EVs contribute to intercellular talk, neural trophic support, myelination adjustment, synaptic plasticity, antigen presentation, etc. [47]. Proteomic studies showed that neuron-specific markers, such as vesicle-associated membrane protein 2 (VAMP2), a constituent of the presynaptic exocytotic machinery, enolase 2, transmembrane protein 132D, a marker for oligodendrocytes and other microglia-specific proteins are present in isolated CSF EVs [70]. EVs may also carry numerous toxic proteins such as prions, α-synuclein or tau, leading to the development and progression of neurological diseases such as Parkinson’s and Alzheimer’s [63]. An additional role for EVs in the CNS could be involvement in the removal of surplus protein, a function that may be particularly relevant to neurodegeneration [70]. Consequently, CNS-derived EVs can be obtained from systematic circulation and considered as potential biomarkers of CNS disorders and their progression [71]. The brain microvascular endothelial cells, astrocytes, and pericytes create the blood–brain barrier (BBB) which is a highly selective barrier regulating the transport between the periphery and CNS [72]. It is considered an obstacle for drug delivery to CNS and most of the small and large molecules including drugs, recombinant proteins, and monoclonal antibodies are not able to go through the BBB [73]. EVs are applicable elements with their prominent characteristic of crossing the BBB. Various mechanisms are involved in this process depending on their cellular origin and can be influenced by neuroimmune conditions [71]. EVs more likely apply the same mechanisms used by viruses to cross the BBB. Studies have demonstrated that the endocytic process is a contributor to EVs crossing the BBB [74]. Given the available knowledge, it is suggested that transcytosis mechanisms used by EVs to cross the BBB are based on crossing cells through vesicles similarly to adsorptive transcytosis [75]. This kind of mechanism can be seen in use by immune cells, viruses and nanoparticles to transit the BBB. Some particular glycoproteins such as wheatgerm agglutinin and HIV-1virus are engaged in this kind of traversal, and mannose-6 phosphate receptor can also mediate it [76,77]. It is noteworthy that during inflammation, transit over the BBB may change due to its disruption [78]. It is promising that expected agents such as drugs, nucleic acids and other large molecules can pass through the BBB via EVs. Systemic injection of exosomes has been found to be effective in the transfer of small interfering RNA (siRNA) into the mice brain [79]. Given their implications in both physiological and pathological processes, it can be helpful to establish beneficial preventive plans and design successful therapeutic alternatives, either to mimic EVs or engineering them for a new purpose [27,80].

## 5. Ageing, CS and EVs

Aging is a well-known process that leads to an increased risk of several chronic diseases including neurodegenerative diseases [81]. Some cellular and molecular hallmarks that contribute to the process of aging are proposed, such as genomic instability, telomere attrition, epigenetic alterations, altered intercellular communication, deregulated nutrient sensing, mitochondrial dysfunction, cellular senescence, stem cell exhaustion and loss of proteostasis [82]. Dysregulation in proteostasis and complexities in protein conformation are the most relevant characteristics of aging [83,84]. The efficiency of the chaperone systems decreases in aged cells which leads to the accumulation of detrimental aggregates of misfolded proteins [28,85,86]. The initial level of molecular chaperone Hsp70 has been described as reduced in cerebral cortex tissue in aged mice versus adult animals [87]. Furthermore, a human study revealed that the ability to enhance Hsp70 expression in the stress context is typically diminished in lymphoblasts with age [88]. Post-mitotic somatic tissues deteriorate with time in aged organisms, making them particularly vulnerable to age-dependent protein aggregation disorders. Ectopic somatic overexpression of CCT8, the subunit that facilitates the assembly of CCT in mammalian cells and that is widely expressed in somatic tissues, caused an increase in lifespan of up to 20% under normal conditions at 20 °C and, surprisingly, up to 40% at 25 °C in C. elegans compared to the control. Likewise, CCT2 overexpression is reported to enhance the lifespan, albeit by a lower percentage. Considering that heat shock disrupts protein structure and promotes the accumulation of misfolded proteins, it can be implied that CCT8 and CCT2 help organisms to live longer through stabilizing protein homeostasis during aging [89]. Due to a drastic imbalance between excessive amounts of protein aggregates and available molecular chaperones during aging, both expression levels and the proper protein folding capability of CS may be altered by the toxicity of accumulated misfolded proteins as a common molecular pathway in numerous human diseases [90,91,92,93,94]. The aforementioned process may be accelerated by various cellular stressors that can prematurely stimulate the same senescence phenotype as seen in senescence induced by telomere shortening [95]. The senescence process is recognized by the Senescence-Associated Secretory Phenotype (SASP), which is responsible for transmitting the signal from the senescent cells to the surrounding tissue [96]. It has been shown that the induction of senescence in human astrocytes in vitro via the extracellular tau or estradiol treatment leads to the secretion of the SAPS factors such as IL-6, IL-1β, TNF-α and IL-8, as observed in the hypothalamus of aged mice [97,98]. Exosomes represent SASP components and highlight their role in mediating cell-to-cell transmission of senescence signals [99]. Although the role of exosomes in the healthy aged brain and in neurodegenerative disease development is still not clear and rather controversial, they are frequently described as feasible carriers of pathogenic misfolded proteins [100,101]. Exosomes have been suggested to play a dual role in promoting the development of neurodegeneration and to prevent spreading of the disease and maintaining normal physiology [102,103]. Exosomes isolated from young mice caused [104] significant downregulation of aging-associated signaling molecules such as insulin-like growth factor 1 receptor (IGF1R) and up-regulate telomerase-related genes in aged mice [104]. Neural derived exosomes are involved in the transport of pathogenic proteins such as phosphorylated tau P-181, Beta Amyloid 42 (Aβ1–42), chatepsin D, repressor element 1-silencing transcription factor (REST) and neurogranin and in aging [105]. Considering that EVs are powerful intercellular communicators, a notable contribution can be assigned to the development of age-related disorders or senescence processes, as some distinctive alternations have been observed in the production of EVs in senescent cells [106,107]. Caveolin-1 and charged multi-vesicular bodies protein 4C (CHMP4C) are the main endosomal regulator genes which are up-regulated by p53 tumor-suppressive pathway [108]. Their expression promotes endocytosis and internalization of surface receptors, including EGFR [105], and improves MVB generation as a part of the ESCRT-III complex, respectively [109]. Furthermore, the over-expression of essential genes required in EV secretion, most impressively tumor suppressor-activated pathway 6 (TSAP6), is attributed to p53 activation. Given that, activation of the p53 during senescence may stimulate the over-secretion of EVs, which signifies the progressive feature of the aging phenotype from the cellular to organismal levels [110]. Investigation of the role of the released miRNAs from senescence cell-derived exosomes into the extracellular environment has uncovered that micro-RNA (miRNA) such as miR-433, miR-34a, and miR-29 can improve the induction of senescence in ovarian and colon cancer cells, respectively [111]. Furthermore, miR-146 is known for senescence-associated inflammation derived from senescent human fibroblasts [112]. Several miRNAs have been identified also in the neural derived exosomes. Analysis of miRNA cargo of neural-derived EVs (NDEVs) in 40 Alzheimer patients showed a statistically significant increase in the content of miR-23a-3p, miR-223-3p, miR-100-3p and miR-190-5p compared with the control group [113,114]. In addition to miRNAs, other types of nucleic acids such as long non-coding RNAs (lncRNAs), DNA, mRNA and circular RNAs (circRNAs) or piwi-interacting RNA (piRNA) were found in the NDEVs; their association with neurodegenerative etiopathogenesis requires further research [102,115]. The neuronal long noncoding RNAs (lncRNAs) are involved in controlling telomere length, epigenetic gene expression, proteostasis, stem cell pool, cell proliferation and senescence, intercellular communication and local regulation of the stability of protein-coding mRNAs that participate in synaptic plasticity [116]. Dysfunctional activity during all the above processes can promote the development of age-related phenotypes [117]. Moreover, upregulation of lncRNA *17A* was seen in the cerebral tissues derived from Alzheimer patients and in the response to inflammatory stimuli such as IL-1α (SASP cytokines) [118]. As the modulator of proteostasis, the lncRNAs contribute to the protein synthesis, trafficking, assembly, degradation and autophagy that are noticed in aging [119]. The lncRNA such as HULC, MEG3, and 7SL modulate autophagy [120,121], while others such as Uchl1 and LncRNA-p21 regulate protein synthesis and degradation through the ubiquitin proteasome pathway [122,123]. Proteostasis declines with age and aberrant proteostasis is one of the major factors of age-related diseases such as Alzheimer’s, Parkinson’s, and Huntington’s disease [83]. 

## 6. Alzheimer’s Disease, CS and EVs

Alzheimer’s disease (AD) is recognized by the accumulation of misfolded amyloid β (Aβ) and tau proteins as amyloid plaques inside or outside of neurons, and leads to the initiation of the neurodegenerative process in the brain [124,125]. Conformational alternation of the tau protein may occur due to its hyperphosphorylation and its elevated aggregation leads to destabilization of the microtubules, which accelerates the neurodegeneration process [126]. Although most cases of AD show a delayed onset at around 65 years old, a small number of patients, about 5%, show an early start of AD due to genetic elements [127,128,129]. Short memory is the main complication of the disease, known as Dementia [129]. During AD, neurons are the most vulnerable cells to apoptosis in the brain tissue as plaques activate the death enzymes in neurons, representatively caspase6, an apoptotic cysteine protease, that results in the disease’s development and neurodegeneration [130]. Furthermore, the expression of brain-derived neurotrophic factor (BDNF) as the principal factor in synaptic plasticity and neuronal survival is down-regulated in AD [131]. Other complexities with Alzheimer’s brains are reported as the malfunction of adhesion molecules involved in synapses such as cell surface [132] and low levels of neurotransmitters, including acetylcholine, catecholamine, glutamate and serotonin [129,133]. Interestingly, the substantial role of molecular chaperones, typically Hsps, has been indicated to inhibit amyloidogenic protein aggregates such as Aβ and tau, and their degradation of those aggregations or misfolded proteins has been indicated in different studies [134]. It is suggested that the clearance of toxic tau and Aβ aggregates is particularly modulated by Hsp40 and Hsp70 through multiple mechanisms [135]. Hyperphosphorylated tau aggregates might be captured by Hsps to block the formation of oligomer or higher order structures and to adjust their degradation through the ubiquitin proteasome and autophagic lysosomal pathways [136]. The human Hsp70 disaggregation machinery (HSC70, DNAJB1, HSPA4) disassembles a variety of pathological amyloid tau aggregates, ranging from fibrils formed in vitro to aggregates recovered from a cell culture model to brain material from AD patients. Despite the fact that the liberated Tau species were mostly monomers, they were seeding-competent and caused self-propagating Tau aggregates in a cell culture model [137]. This motivates the idea that chaperone-mediated disaggregation stimulates the prion-like distribution of amyloid tau aggregates in vivo, subsequently resulting in amyloid overload [138]. Distribution of tau as a cytoplasmic protein that is deposited intracellularly involves the release and uptake of seeding-competent from the cytosol of the donor and receiving cells, respectfully [139]. HSC70, together with its co-chaperone DNAJC5, promotes tau release into the extracellular space in cell culture and in a mouse model [140]. Furthermore, Hsp90 cooperatively with its cochaperones showed an imperative role in refolding denatured or misfolded tau and Aβ proteins [141]. Despite what was mentioned above, an age-dependent decline in cellular quality control, impaired function, and decreased levels of different molecular chaperones may be the main contribution to the induction of pathogenic conditions as observed in various neurodegenerative diseases [142,143]. In this sense, in recent studies, reduced levels of mRNA expression of the CCT complex associated with the AD brain have been demonstrated. Since tau protein folding, stabilization, and degradation have been shown to be controlled by CCT complex along with many other molecular chaperons, such as Hsp90 and Hsp70; these findings of insufficient competency can be considered influential in neurodegenerative diseases [144]. On the other hand, there is the considerable idea of EVs’ implication in spreading AD that endosomes are involved in b-cleavage and then exosomes carry about 1% of Aβ to the extracellular space. It is noteworthy that amyloid plaques are abundant in exosome-associated proteins such as flotillins and Alix, which indicates the participation of exosome-associated Aβ in plaque formation [145]. Although low levels of Aβ are secreted through exosomes, it can gradually promote the disease, as well as what is happening with prion disease [146]. Further studies indicated the role of EVs as a peculiar means of amyloid precursor protein (APP) secretion and AD progression [147,148]. EVs obtained from the brains of transgenic mice showed an increase in APP and Aβ compared to wild type mice [149]. EVs contain secretase and proteases that contribute to the amyloidogenic cleavage of APP inside them, suggesting EVs as applicable diagnostic and therapeutic factors [147]. In an interesting study, it is claimed that chloroquine treatment, a vesicle trafficking inhibitor, improved exosome-associated GM1 ganglioside, which is an accurate site for amyloid β-protein binding. Exogenous Aβ can proceed to fibrillogenesis via GM1 on the exosomal surface. Consequently, exosomal GM1 caused by disruption in the endocytic pathway function is determined as a potent pathway of Aβ accumulation in Alzheimer’s-affected brains [150]. The researchers showed that PAR-4/ceramide-enriched exosomes triggered apoptosis on primary cultured astrocytes. Prostate apoptosis response-4 (PAR-4) is a protein that makes cells sensitive to the sphingolipid ceramide and the amyloid peptide induced the secretion of PAR-4 / ceramide-enriched exosomes, while astrocytes with deficient neutral sphingomyelinase 2 (nSMase2) did not undergo apoptosis, implying the role of the ceramide generated by nSMase2 for amyloid-induced apoptosis [151]. Since nSMase2 inhibited the EVs release and prevented amyloid deposition and plaque formation in the brain it was characterized as a promising agent in Alzheimer therapy [152]. EVs in CSF of Alzheimer patients have been found to be significantly enriched in myeloid protein compared to healthy controls that improve neurodegeneration. In particular, atrophy of the hippocampus and activated microglia leads to enhanced EV release in patients and was associated with massive neuronal death [153]. These findings indicated that the lipid composition of microglial derived EVs such as sphingomyelin, cholesterol, ceramide, flotillin-2, GM1 and GM3 gangliosides can help to uptake the extracellular insoluble aggregates to construct the neurotoxic soluble pattern of Aβ [154]. EVs are also assigned to show protective characteristics in AD. In this regard, immunoproteomic analysis revealed that EVs released from mouse primary neurons contain cystatin C which is a potential neuroprotective agent in AD and low detection of which is considered as a diagnostic factor [155,156]. Moreover, administration of siRNA loaded exosomes targeting BACE1 has been reported as an applicable approach to modulate β-amyloid accumulation during AD [79]. It should be noticed that exosomes can promote the production of Aβ and enhance microglial uptake of Aβ to be degraded by lysosomes [157]. Since neuronal exosomes are enriched in glycosphingolipids (GSLs), it makes them more potent to bind to amyloids on their surface. In addition, Aβ and amyloid depositions decreased in the brains of APP transgenic mice after the delivery of neuronal exosomes [158,159]. Furthermore, isolated EVs from blood and CSF of Alzheimer’s patients showed abundant phosphorylated tau protein [160,161], and it has been discovered that EVs are the major carriers for tau protein transportation into the extracellular environment, leading to tau aggregation and distribution of the disease [162,163]. Human CSF-derived EVs contain phosphorylated tau protein, indicating a common biomarker for AD [161]. Exosomes derived from the brain of PS19 mice have been shown to deliver tau protein to neuronal cell cultures [164]. In particular, it has been reported that although tau contributes to both Alzheimer and Parkinson diseases, the transportation process of tau is different in these two disorders, and CNS-derived EVs carrying tau are better considered a biomarker of Parkinson’s disease than AD [165]. Taking into account the insufficient symptoms at the beginning of the disease and the fundamental role in diagnosis, it seems necessary to design further works to characterize EVs as early biomarkers for AD to find vulnerable patients before massive neuronal loss in their brain [166].

## 7. Parkinson Disease, CS and EVs

Parkinson’s disease (PD) is identified as the second-most common progressive neurodegenerative complication. It is considered a chronic disorder which is clinically manifested via hypokinesia, rest tremor, rigidity and postural inconsistency. The basic pathological features of PD consist of the vast degeneration of dopaminergic neurons in the brain’s basal ganglia, aggregation of Lewy bodies (LB) and accumulation of the α-synuclein protein in numerous of the remaining neurons. However, it seems that some of the α-synuclein oligomers cause neurotoxic effects of these protein aggregations [167]. In fact, the defective activity of molecular chaperones impedes the clearance of these massive protein aggregates in aged neurons, promoting toxicity and ultimately resulting in the pathogenesis of PD [168]. Recent studies have indicated a particular interaction between eukaryotic molecular chaperonin CCT and soluble oligomers of α-syn A53T to prevent their fatal effects and the formation of amyloid fibers. A53T is known as a point mutation leading to oligomerization and develop amyloid fibrils in early-onset PD. Evidently, in chaperon system impairment, extracellular α-syn oligomers are recognized as extremely neurotoxic when taken by neuronal cells and result in pathogenesis [169,170]. It is noteworthy that, although pluripotent stem cells are rich in CCT complexes, their expression levels are decreased in differentiated cells. Given that, mature neurons are desperately prone to misfolded protein toxicity. Therefore, stimulating the expression of CCT subunits to form the complex may recover the proteostasis potency of adult cells as a probable neurotherapeutic approach [171]. Investigations in PD brains indicated lower levels of Hsc70 in the substantia nigra pars compacta and amygdala, the main anatomical locations for α-synuclein aggregates and LB, compared to AD brains or healthy aged control cases, suggesting impaired chaperone-mediated autophagy in PD [170]. In contrast, cells respond to protein aggregation through up-regulation of molecular chaperones such as Hsp90, which is detectable in patients with LB and Lewy neurites in PD [172]. The correlation of Hsp90 with α-synuclein in under oxidative stress conditions suggests that Hsp90, like Hsp70, contributes to the removal of protein aggregate [173]. Furthermore, elevated levels of Hsp70, Hsp40 and Hsp27 have been seen following overexpression of α-synuclein in a mouse model [174,175]. In addition, the disrupted function of α–crystallin B, a member of a small chaperone family, has been shown through interaction with α-synuclein in LB [176]. However, in the age context, diminished proficiency of Hsps to monitor protein aggregation and degradation may allow the gradual accumulation of misfolded proteins, which consecutively threatens the ability of molecular chaperones to control cellular protein homeostasis and triggers the neurodegenerative process [177]. While the pathologic effects of the prefibrillar species have not been successfully described yet, in a similar way to tau and Aβ protein distribution, EVs are their effective carriers and functional mediators of toxic α-synuclein aggregation and propagation between neurons, thus promoting the PD [178]. Potentially, target cells receive α-synuclein carried by EVs more efficiently producing toxic effects than free α-synuclein oligomers. Since EVs represent α-synuclein oligomers both on their outside and inside, they play a crucial role in the progression of Parkinson’s disease [179,180]. Furthermore, EVs circulating in blood and CSF circulating EVs of patients with PD have been found to be highly enriched with α-synuclein and are remarkably correlated with the stage of the disease [181]. EV-associated α-synuclein induces cytotoxicity in the recipient cells via enhancement of LB in different parts of the brain [182,183]. Evidently, it is suggested that exosomal gangliosides can modulate the catalytic environment and stimulate the aggregation of α-synuclein [183]. It has been demonstrated that P-type ATPase ion pump (PARK9/ATP13A2) regulates either the biogenesis or secretion of exosomal α-synuclein. It has been noticed that knock-down of PARK9 reduced the release of exosomal α-synuclein, and on the other hand, secretion of α-synuclein via exosomes increased following overexpression of PARK9 [184,185]. Another acceptable molecular mechanism may be disruption in the autophagy lysosomal pathway (ALP), resulting in insufficient elimination in lysosomes, suggesting the adaptive secretion of exosomal α-synuclein [180,186,187]. A significant correlation has been found between the efficacy of the cell protein degradation system and level of α-synuclein content of EVs, since they used bafilomycin A1, a pharmacological inhibitor, in SH-SY5Y cells that altered ALP and prevented the fusion between autophagosomes and lysosomes that caused exosomal α-synuclein secretion to accelerate [185,188,189]. Glial cells such as [91,93] microglia also have a regulatory role in EV activity during PD. Upon exposure of BV-2 microglia to α-synuclein, activated exosomes containing high levels of MHC class II and membrane TNF-α were harvested. Neuronal death was observed following the treatment of cortical neurons with those activated exosomes in rats [179]. The evaluation of CSF content of exosomal α-synuclein may be a promising diagnostic biomarker for PD. EVs isolated from PD patients CSF trigger soluble α-synuclein oligomerization in a dose-dependent manner in the recipient cells [190]. On the other hand, EVs released by monocytes and macrophages were applied in a therapeutic approach to improve the efficiency of loaded PD drug delivery to target cells and reduce trapping in the mononuclear phagocyte system [191]. Remarkable neuroprotective effects were reported on the exosomal-based delivery of glial cell-derived neurotrophic factor (GDNF) in models of PD therapy in vitro and in vivo [192,193]. Finally, it seems that EVs are potent structures both in propagating PD and the prevention of its pathology.

## 8. Prion Disease, CS and EVs 

In prion diseases, such as AD and PD, misfolded proteins contribute to fatal transmissible neurodegenerative pathology, including Fatal familial insomnia (FFI), Creutzfeldt-Jakob disease (CJD), Gerstmann–Sträussler–Scheinker disease (GSS) and Kuru. These develop as genetic, infectious or sporadic disorders, and they are mostly manifested during old age. Complexities such as motor dysfunction, dementia and cognitive deterioration, spongiosis, astrogliosis and cerebral deposition of insoluble prion proteins (PrP^C^) are observed in prion diseases [194,195]. The PrP^C^ is present in either neuronal or glial cells as a glycosylphosphatidylinositol (GPI)-linked extracellular membrane protein. There is still no accurate understanding of its biological function, while several cellular activities are attributed within the CNS, such as synapse formation and neurite growth [196]. In the context of prion diseases, the original form of the prion protein PrP^C^ is transformed into the pathologic isoform PrP^Sc^, which is more potent in aggregation and resistance to proteases [197]. The steady accumulation of misfolded proteins disrupts homeostasis, prompting cellular stress responses including elevated protein chaperone levels, which start as a pro-survival mechanism aimed at restoring protein synthesis and degradation balance, but under chronic stress, activation of apoptosis is inevitable [197]. In vitro and in vivo investigations of the involvement of the molecular chaperone Hsp70 in chronic prion infection of mammalian cell contexts indicated that the heat shock response considerably reduced PrP^Sc^ accumulation and its replication. More notably, in Hsp70-deficient mice, the progression of prion disease was boosted relative to WT mice and high levels of Hsp70 levels have been reported in patients with CJD [198] and scrapie-infected mice [199]. In contrast, while Hsp104 is essential for the generation of all yeast prions, deficient Hsp104 caused the removal of all amyloid-based prions in the yeast. It cleaves the aggregated filaments by means of its chaperone disaggregating activity while collaborating with Hsp70 and Hsp40, and as a consequence, it releases the monomers to produce new prion seeds. Interestingly, overexpression of Hsp104 can also control PSI+, prion of Sup35p that is a component of the translation termination system, a type of prion disease [200]. To achieve this goal, it requires critical activity of the co-chaperone Sti1p, a protein that interacts with Hsp70, Hsp90, or Hsp104 chaperones. [201]. Although activating cellular stress mechanisms to recover proteostasis during prion diseases seems promising, such cellular responses have faded with age, since the accumulation of PrP^C^ aggregates was continuously increased in transgenic drosophila overexpressing PrP^C^ with age [202,203].

Although in vivo interaction of PrP with Hsp60 may seem unlikely, it has been declared that only this chaperonin can trigger the aggregation of recombinant PrP de novo, without the presence of a template of pre-existing PrP aggregates. It is mainly confirmed in a mildly acidic pH of 5.5, such as what is seen in the endosomal compartment where PrP^Sc^ synthesis is hypothesized to develop [204]. These findings can imply that during prion disease, some conditions may exist in which a molecular chaperone or chaperone-like interaction partner of PrP converts PrP^C^ to PrP^Sc^ [205]. An active interaction between the monomeric, oligomeric, and fibrillar forms of prion protein (PrP) and chaperonin CCT has been reported, while glycation could diminish prion protein affinity for the CCT complex. Interestingly, CCT binding to the recombinant ovine prion protein stimulates its aggregation and the development of fibrillar structures under certain conditions. It is worth noting that any defect in CCT function due to ATP deficiency, blockage by misfolded proteins, aging, etc. may improve its capacity to build active fibrillar structures and amyloidogenic proteins rather than reducing their breakdown [206,207]. Given that, a harmony between free and chaperone-associated forms of amyloid proteins is necessary in the cell, since their accumulation due to dysfunction of the chaperone system leads to disease [207]. On the other hand, EVs are considered to have a substantial role in the pathogenesis and development of prion diseases, since PRNP (prion proteins) has been associated with isolated EVs from CSF and blood [208]. It has been reported that EVs are the major conveyers of PRNP transmitters for the RK13 cell line [146]. Inhibition of EV secretion in various PRNP-expressing cell lines, using pharmacological agents, can prevent intercellular PRNP trafficking. Neutral sphingomyelinase has been found to be the main mediator for the conduction of PRNP into the EVs [209]. EVs derived from the PRNP-affected neuronal cell line (GT1-7) are functional agents in prion transfer to neurons and even other non-neuronal cells [210]. Isolated EVs from human tissues of prion disease contain abundant levels of miRNAs such as miR-146a, miR-103, miR-125a-5p, miR-342-3p and let-7b, emphasizing the crucial role of EV content in disease promotion besides the propagation of PRNP [211]. However, extensive investigations seem essential to explore the distinct role of EVs in prion pathology. Since the RK13 cell line has been demonstrated to release numerous types of PRNP applying various possible cellular mechanisms [212], achieving first knowledge of EV functions in prion diseases can be beneficial to analyze the different mechanisms that support disease pathogenesis [163].

## 9. Huntington’s Disease, CS and EVs

Huntington’s disease (HD) is an incredible progressive and lethal neurodegenerative state that represents characteristics such as cognitive and behavioral degradation and motor dysfunction. It particularly appears during the 4th decade of life; however, there is variation from 20 to 65 years. Unfortunately, there are still no successful cures for preventing neurodegeneration. For example, mutation in the huntingtin (HTT) coding gene on chromosome 4 involving repeat CAG is responsible for HD as an autosomal dominant disease. It is well known that the mutant huntingtin protein causes damage to intracellular Ca2+ homeostasis, disrupts intracellular trafficking, and impairs gene transcription [213,214]. It has been suggested that the polyglutamine (poly Q) repeat of mutant HTT and expanded repeat RNA are modulators of neurotoxicity in the pathogenesis of HD. In fact, neurodegeneration has developed through induction of homo/heterodimerization caused by mutant HTT, leading to protein aggregation and fibril formation [215,216]. The potency to block protein aggregates and suppress neurodegeneration, including HD, has been indicated through various animal model studies; however, this characteristic of the chaperone system is a matter of alternation with aging [217]. Hsp70 is involved in HD since it plays a significant role in preventing the toxic effects of accumulated protein aggregates, while it could not suppress their formation in mouse and *Drosophila melanogaster* models [218,219]. Furthermore, when two chaperones Hsp70 and Hsp40 cooperate, they function more effectively as inhibitors of aggregate formation in polyQ diseases, including HD. Their overexpression showed reduced production of large aggregates by promoting the formation of smaller detergent-soluble aggregates [220]. Studies on HD mammalian cell models have shown that yeast heat shock protein Hsp104 has the ability to decrease aggregate formation and prevent cell death. Furthermore, transgenic HD mice overexpressing Hsp104 revealed less aggregate formation as well as an increased survival rate by 20% compared to controls [221]. In addition, the CCT complex has been found to be capable of reducing HTT aggregation in the cell culture model. Apical domain involvement of CCT with the mutant HTT fibrils to encapsulate oligomers has been demonstrated by cryoelectron tomography [222]. CCT chaperonin subunits including CCTs 1, 2, and 4–6 and bovine overexpressed subunits CCT1 and CCT4 contribute to the suppression of polyglutamine aggregation in Caenorhabditis elegans and yeast, respectfully [223], [18], [224]. Recent in vitro studies described the ability of human CCT5 subunit to form a homo-oligomeric complex that is made up of 16 identical CCT5 subunits constituting a double-ring barrel shape, structurally similar to the CCT complex with the potential to inhibit HTT aggregation [225]. The chaperone system showed loss of function in old rats when compared to younger ones in an animal model of aging [226]. There may be different mechanisms that explain defective chaperone activity in older animals, including lower levels of Hsp90, being particularly affected by oxidative stress [227], interrupted balance between free chaperones and abundant misfolded proteins [91]. These findings would suggest ineffective and “sick chaperones” which will be unable to fulfil the increased folding needs of damaged, misfolded proteins in aging cells [94].

Evaluation of the poly Q HTT protein, expanded repeat CAG RNA or normal protein transmission capability of the EVs to the neurons of striatum in vitro showed the integration of expanded repeat RNA and poly Q protein presented within EVs during neurodegeneration, suggesting biomarkers that are applicable and promising objectives for HD disease [215]. EVs have been described as conceivable agents to deliver hydrophobically modified siRNAs (hsiRNAs) to enhance cellular internalization and improve CNS diffusion into mouse striatum to reduce expression by up to 35% of HTT mRNA [228]. In addition, exosomes derived from adipose stem cells are potent to inhibit apoptosis by reducing mutant HTT accumulation and improve the mitochondrial function in HD in vitro [229,230]. The overexpression of REST, the target gene of miRNA-124, occurs following low levels of miRNA-124 resulting in the suppression of genes such as brain-derived neurotrophic factor (BDNF) while the delivery of miRNA-124-loaded EVs attenuated the expression of the REST gene in striatum and stimulates neurogenesis [231]. Given that, new feasible EV-based approaches can be designed to treat HD and other neurodegenerative diseases.

## 10. Therapeutic Applications of EVs and Molecular Chaperones in CNS Disorders

Most neurodegenerative disorders are induced by the toxicity of protein aggregates, which results in the loss of neurons [232]. Since efficient function of the proteins demands proper folding, an appreciable therapeutic approach to improve cell survival could involve strategies to inhibit protein misfolding and aggregation [233]. However, various pathways are involved in the maintenance of proteostasis within the cells, but any defect in the proteins’ proper folding process can cause strict neurodegenerative states which subsequently develop into apoptosis and cell death [234]. Molecular chaperones, as essential mediators of protein homeostasis, could be a promising form of therapy in proteinopathies, including neurodegenerative diseases [230]. In fact, many molecular chaperones, in particular DNAJ chaperones family members [235], were determined to decrease polyQ aggregation efficiently and further to prolong the onset of disease in vitro in cell models, and in vivo in flies, xenopus and mice HD models [236]. Furthermore, imbalanced chaperones such as Hsp104, Hsp40, Hsp70 and Hsp90 in terms of overexpression or deficiency have been determined to treat yeast prions, which is known as the anti-prion effect of chaperones. “Anti-prion system” describes elements that eliminate prions in healthy cells with no defect or excessive proteins. To clarify that, the prions that develop in the absence of interested chaperone were inhibited when the normal level of the chaperone was reestablished [201]. The complexity of CNS, along with the presence of the BBB, impedes the successful delivery of therapeutic elements via ordinary systematic administration. To convey exogenous healing agents to the CNS, it seems crucial to preserve them from the destructive effects of the disease environment and enzyme digestion, and secure acceptable infiltration into the nervous tissue. Consequently, a substantial and efficient approach will be developed to improve the uptake of biomedical agents in the neurodegenerative context [237]. Encapsulating therapeutics can assure their activity and support their specific efficient uptake. EVs have appeared to be strong candidates as biological carriers for various biomolecules, including proteins [238]. The involvement of EV in the regulation of physiological processes make them an applicable alternative in regenerative medicine, as well as in the immune system [239]. Intentional interference to decrease the role of EVs in disease propagation, utilizing their inherent therapeutic potential or manipulating them as a drug delivery route are the main developing areas in EV therapy [240]. Being capable of passing the BBB, the potential to target specific cell types and cargo conveyance, the insusceptibility to immune reactions and the credible preservation of their bioactive contents and drugs are promising therapeutic characteristics of the application of EVs to CNS disorders [27,241]. Beneficial effects of EVs are assigned to their potent cargoes, such as growth factors, proteins, bioactive lipids, microRNA, mRNA and noncoding RNA [242,243]. 

EVs from Hsp40- and Hsp70-overexpressing cells repaired proteostasis and decreased protein aggregates in the adjacent cells in the brain of the mice and drosophila containing those molecular chaperones [244]. Furthermore, astrocytes that express higher levels of DNAJB6 chaperone could suppress the accumulation of protein aggregates in the neurons of drosophila [245]. Moreover, a co-culture of overexpressing DNAJB6-WT cells reduced the accumulation of Q74-GFP, a susceptible protein to create insoluble aggregates, in neurons, while the decline in polyQ aggregation was not seen through co-culture of the upregulated nonfunctional mutant DNAJB6-M3cells. These findings may support the idea of EV-mediated chaperone transportation to restore protein homeostasis in protein misfolding pathologies, including neurodegenerative diseases [245,246]. In addition, neural stem cells (NSCs)-derived exosomes bearing DNAJB6 acted as anti-amyloidogenic elements, decreased HTT aggregates in the cells containing excessive polyQ and also in the HD mice brains and further postponed the onset of HD [245]. EVs also carry small heat shock proteins families (HSPBs) loaded in their lumen or attached to their membrane in the extracellular environment, and which could be taken in by the target cells [247]. Once connected to the cell surface receptors in the extracellular zone, HSPBs act as signaling molecules that indicate an inflammatory or immune reactivity condition due to any cellular injury [248,249]. Extracellular HSPB1 can be considered a pro-inflammatory factor and immune response modulator by stimulating nuclear factor-kappa beta (NF-κβ), release of inflammatory reagents such as IL-6, and IL-8 [250], and growth factors upon interaction with surface receptors which is generally seen in neurodegeneration. Moreover, extracellular HSPB1 showed chaperone characteristics by being able to bind to the extracellular Aβ in the Alzheimer brains [251], and α-synuclein fibrils, which are found in Parkinson’s disease, to inhibit their expansion and protect against their toxic effects [252]. The proven therapeutic effects of HSPB1 in neurodegenerative diseases require evaluating the specific roles of the other members of HSPB members in neurodegeneration as future practical treatments [234]. Although chaperone systems mainly represent honorable outcomes, some of their activities may still have disadvantages. In that case, Hsp70 can function constructively or in a deleterious way. When the amyloid aggregates, it can cause the liberation of monomers which are more toxic, seeding and spreading-competent species that promote prion-like propagation [138]. Furthermore, the failure in chaperone-mediated delivery to the ubiquitin-proteasome system (UPS) or autophagy to eliminate misfolded proteins may generate seeding- and spreading-competent subunits of sick proteins, or, in the case of autophagy, accelerate their delivery to surrounding cells. The proteostasis functions considerably strong for young humans to disaggregate, refold or destroy the misfolded proteins, but this ability is progressively weakened in elderly individuals in whom the removal of the disaggregated substrates has been found inefficient. Consequently, chaperones may indirectly assist dispersion, as they are also involved in trafficking pathways, which are associated with the transmission of prion-like proteins between cells [253].

EVs are appropriate in neurodegenerative conditions; in particular, natural or artificial EVs containing molecular chaperones have received a great deal of attention since they possess the ability to modulate proteostasis and the immune response [254]. 

It also seems possible that a link between cerebral ischemia, chaperones, and extracellular vesicles may exist. It is well known that brain ischemia (due to several reasons such as cardiac arrest, shock, carotid occlusion, hypotension, asphyxia, or anemia) may contribute to the development of Alzheimer’s disease [255]. On the other hand, it was shown that the exosomes of neurons act as an endogenous post-ischemic protective factor by inhibiting microglial phagocytosis and thus mitigating ischemia-induced neuronal death [256]. This raises the question of whether any therapeutic application of chaperone-containing exosomes can be effective in preventing and treatment of Alzheimer’s disease. [257]. This so far remains unanswered but there are several studies that have suggested that exosomes derived from multipluripotent mesenchymal stromal cells play a neuroprotective role in traumatic brain injury and neurodegenerative disorders [258,259].

Various strategies include trying to expand the therapeutics application of EVs including as a drug carrier, manipulating EV producer cells genetically or some adjustments on the EVs themselves in order to incorporate therapeutic agents in their lipid membrane or charging into their hydrous core [260,261]. The notion of EVs for drug delivery was determined in wild-type mice to study Alzheimer’s. The modified DC-derived EVs to express Lysosome-associated membrane protein 2 (LAMP2) and integrate with the neuron-specified rabies virus glycoprotein (RVG) peptide, transferred the exogenous siRNA to neurons, microglia, oligodendrocytes and consequently, suppressed the Beta-Secretase 1 (BACE1) in both mRNA and protein levels in the brain [79]. Therapeutic properties of EVs, exclusively for RNAi-delivery, are remarkably emphasized by the recognition of peptides with specific affinity to bind to the particular cell types in the brain [262]. While EVs are undoubtedly key biological players in intercellular communication with an appreciable therapeutic potential, their precise mechanisms and characteristics still stand as great issues of discussion. In this regard, the amounts of favorable agents such as small RNAs within different types of EVs and engineering their packaging and actions are among the main subjects under study [263,264]. Obviously, competent procedures are required to manage exogenous agent loading into EVs as well as cell culture modifications to generate EVs, efficient purification and characterization and eventually confirmation of being non-toxic prior to their clinical application [265]. Moreover, the standardization of potent drug loading into EVs and their exact loading method, the assessment of various cell types to harvest effective EVs in order to secure functional drug delivery, specialization in tissue-targeting and crossing biological barriers are subjected to profound investigation [79]. In the end, besides manifestation of their biology, accurate characterization of EVs through multiple methods will prevent the undesirable stimulation of immune reaction and also reduce the transmission of unfavorable genetic factors, proteins or noxious elements such as viruses before being used in clinics [240]. Therapeutic applications of EVs in CNS disorders presented by various studies are summarized in Table 3.

## 11. Conclusions

Neurodegeneration is a destructive state that principally targets neurons in different functional anatomical regions. Neurodegenerative diseases are categorized according to their clinical manifestations, which is how symptoms are associated with the neuropathological findings. Considering the advantage and significance of clinicopathological diagnostic methods, it appears to be more purposeful to identify neurodegenerative diseases based on their molecular pathways of pathogenesis that are even common among some of them. Aberrant protein interactions and typical toxic deposits of protein aggregation that can be either intra or extracellular are the main features of neurodegenerative diseases that imply the necessity of a well-functioning normal chaperone system. EVs are potent biocarriers to convey a variety of pathogenic or therapeutic elements. They play a critical role in the initiation and progression of neurodegenerative diseases and provide new biomarkers and therapeutic goals. Monitoring the fate of misfolded proteins by focusing on chaperone system as well as manipulating EVs production and their content may become striking novel therapies to control neurodegeneration. Given that, effective therapeutic approaches can be designed that are applicable in a number of apparently different neurodegenerative diseases. 

## Figures and Tables

**Figure 1 ijms-24-00927-f001:**
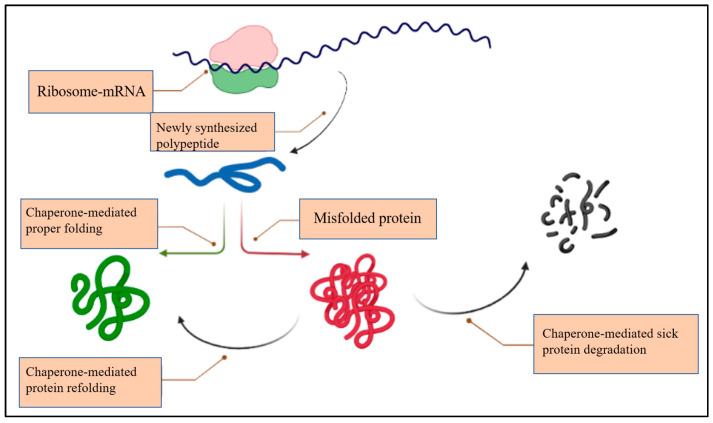
The role of chaperones in the protein-folding process.

**Figure 2 ijms-24-00927-f002:**
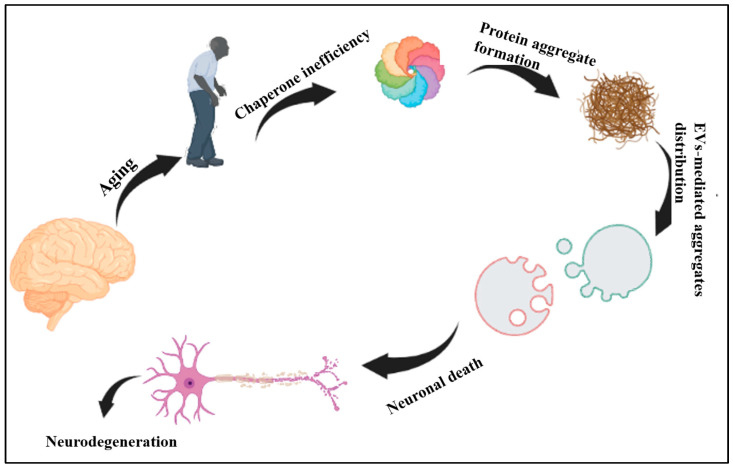
Schematic explanation of the role of chaperones and EVs in the induction and propagation of neurodegeneration.

**Figure 3 ijms-24-00927-f003:**
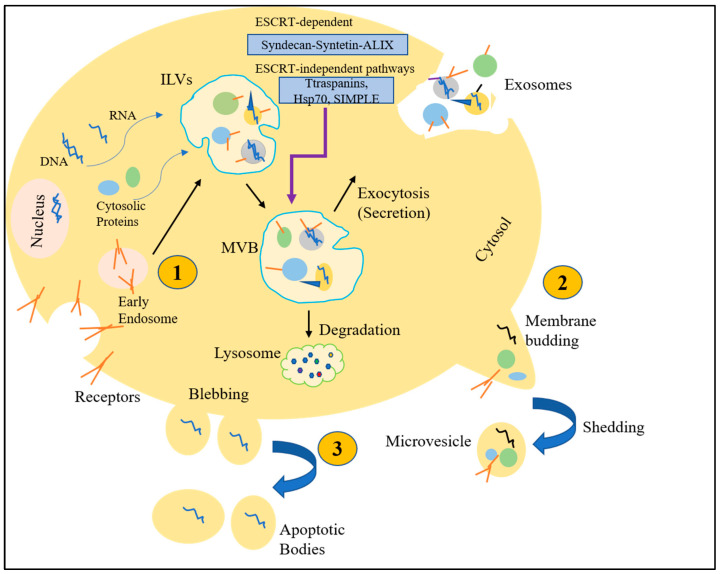
Extracellular vesicles biogenesis and secretion. A eukaryotic cell is schematically represented creating and secreting EVs. (1) Biogenesis of exosomes starts as ILVs following budding into early endosomes. However, ESCRT-dependent or ESCRT-independent MVBs form and undergo one of two fates; degradation through fusion with lysosomes or release of exosomes through fusion with the plasma membrane. (2) Outward budding and fission of the plasma membrane leads to shedding of micro vesicles to extracellular space regulated by phospholipid redistribution and cytoskeletal protein contraction. (3) Upon programmed cell death, apoptotic bodies are formed by actin-myosin mediated membrane blebbing and known as the largest EVs.

**Figure 4 ijms-24-00927-f004:**
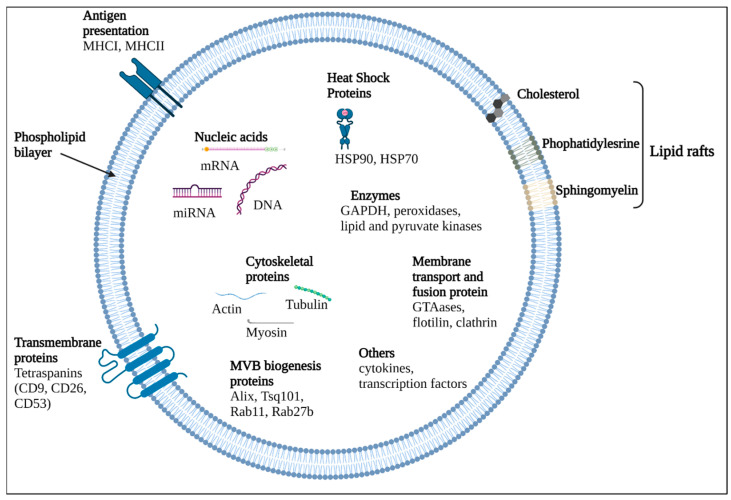
Extracellular vesicle composition. EVs carry various components including nucleic acids (DNA and RNAs), enzymes (preoxidases, lipid and pyruvate kinases) and proteins. EVs bilayer membrane composition includes phospholipids, lipid rafts such as cholesterol, phosphatidylserine and sphingomyelin, but also contains membrane and transmembrane proteins (tetrasapanins, MHCI, MHCII).

**Table 1 ijms-24-00927-t001:** Major molecular chaperones.

Molecular Weight (KDa)	Classical Family
**≥200**	Sacsin
**100–199**	Hsp100–110
**81–99**	Hsp90
**65–80**	Hsp70/DnaK
**55–64**	Hsp60 (chaperonins Group I and II, e.g., Cpn60 and CCT)
**35–54**	Hsp40/DnaJ
**≤34**	sHsp (crystallins)

**Table 2 ijms-24-00927-t002:** Major extracellular vesicles (exosomes and microvesicles).

	**Exosomes**
**Biogenesis**	Generated from multi vesicular bodies (MVBs), following formation of late endosomes, inward budding of the limiting membrane causing intraluminal vesicle (ILVs) formation and accumulation in the lumen, known as MVBs, which may either unite with the lysosomes to degrade their intraluminal cargo or fuse with the plasma membrane to release exosomes into the extracellular space [40]. ESCRT-dependent: Endosomal-Sorting Complex Required for Transport (ESCRT) machinery consists of four protein complexes including ESCRT-0, ESCRT-I, ESCRT-II, ESCRT-III and the associated ones such as VPS4, VTA1, ALIX also called PDCD6IP. Ubiquitinated proteins of the endosomal membrane are required for recognition to start the process [41]. ALIX, a helper component of the ESCRT, assists in the budding and separation process during exosome biogenesis [42]. ESCRT-independent: Tetraspanin molecules (CD9, CD53, CD63, CD81 and CD82) have been shown to play a role during the ESCRT-independent exosome biogenesis pathway [43]. Small integral membrane proteins of the lysosome/late endosome (SIMPLE) are also involved in MVB formation and exosome production in the ESCRT-independent pathway [44].
	**Microvesicles**
	Outward budding and fission of plasma membrane regulated by phospholipid redistribution and cytoskeletal protein contraction as the most common event in MV discharge [45]. Appearance of phosphatidylserines in the outer layer of plasma membrane in association with actin-myosin contraction has been shown during MV formation [46]. Phosphorylation and activation of the myosin light chain kinase is involved in MV secretion [47].
	**Exosomes**
**Secretion**	Transmembrane protein complex soluble N–ethylmaleiamide-sensitive factor attachment protein receptor (SNARE) is involved in exosomes secretion pathway and regulation of the vesicle transport, docking and fusion process [48]. Interaction of the vesicular SNAREs (vSNAREs) localized on MVBs membrane with the target SNAREs (tSNAREs) on the intracellular side of plasma membrane allows membranes fusion to excrete the exosomes [49]. Ras associated binding GTPases proteins, Rab GTPases such as Rab27A, Rab27b, and Rab35 are involved in exosomes secretion [23,50].
	**Microvesicles**
	ADP-ribosylation factor 6 (ARF6) activates the phospholipase D (PLD) that causes the phosphorylation of the extracellular signal-regulated kinase (ERK) on plasma membrane and results in the activation of myosin light chain kinase (MLCK) which is involved in MV secretion [51]. Binding of ESCRT-I subunit tumor susceptibility gene 101 (TSG101) to a tetrapeptide protein within the Arrestin 1 domain–containing protein 1 (ARRDC1) in plasma membrane stimulates MVs secretion [52]. Other external factors including calcium influx through redistribution of the phospholipids and hypoxia lead to stimulating of RAB22A expression and increase MVs secretion [53,54].
	**Exosomes and Microvesicles**
**Uptake**	Interaction of a membrane protein on exosomes with their receptor on the target cell surface [55], e.g., heparan sulfate proteoglycans on the EVs surface, captures fibronectin, which binds to the heparan sulfate on the target cells [56]. Protein cleavage on the EV membrane by proteases and matrix metalloproteinases (MMPs) results in the release of soluble ligands to bind to the receptors of the target cell membranes [57]. Internalization of EVs by the recipient cells to release the content into the target cells [49,56].

**Table 3 ijms-24-00927-t003:** Therapeutic applications of EVs in CNS disorders.

First Author/Year	Target Group	CNS Disorder	Source of EVs	Effective Cargoes	Main Results
Zhang et al. (2020) [266]	Old people mesenchymal stem cells (OMSCs)	Aging	Umbilical mesenchymal stem cells (UMSCs)	miR-136	Reduced senescence phenotype of OMSCs
Mu et al. (2014) [267]	Mice	Aging	Plants (ginger, grapefruit and carrot)	Proteins, lipids, microRNAs	Expression of heme oxygenase-1 (HO-1), IL-10 and Nrf2
Narbute et al. (2019) [268]	Rats	Parkinson	Stem cells from the dental pulp of human exfoliated deciduous teeth (SHEDs)	Proteins, lipids and RNAs	Motor function improvement and normalization of tyrosine hydroxylase in striatum and substantia nigra
Haney et al. (2015) [193]	Mice	Parkinson	Raw 264.7 macrophages	Loaded catalase	Decreased oxidative stress and increase neuronal survival
Yuyama et al. (2014) [158]	Mice	Alzheimer	Neuroblastoma, Neuro2a (N2a) derived exsosomes	Abundant glycosphingolipids (GSLs)	Carrying Aβ on the exosome surface GSLs to deliver it to microglia
Alvarez-Erviti et al. (2011) [79]	Mice	Alzheimer	Dendritic cells (DC)	siRNA	BACE 1 gene knockdown in brain
Zhuang et al. (2011) [241]	Mice	LPS-induced brain inflammation	Tumor cell lines such as 4T1, CT26	Exosomes loaded with curcumin and JSI-124	Induction of apoptosis in microglia and inhibit brain inflammation
Takeuchi et al. (2015) [244]	Neuro2a cultured cells and eyes of Drosophila	Aggregated polyglutamine	Neuro2a cells	Hsp40, 70-rich exosomes	Elevated chaperones, Hsp40 and Hsp70, improves proteostasis both in cultured cells and in Drosophila
Bason et al. (2019) [236]	Mice	Huntington	Human DNAJB6	DNAJB6-rich EVs	Suppression of PolyQ aggregation and related neurode- generation.

## Data Availability

The data presented in this study are included in the article; further inquiries can be directed to the corresponding author.

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
