# Peer review of "Contribution of Extracellular Vesicles and Molecular Chaperones in Age-Related Neurodegenerative Disorders of the CNS"

_ijms, 2023, doi:10.3390/ijms24020927_

Round 1
Reviewer 1 Report
In the present review, the authors provide an overview about the neurodegeneration process caused mainly by alterations in chaperones systems and EVs performance leading to neuropathies. Moreover, authors also outline the potential of EVs as diagnostic biomarkers, as well as in EVs-based therapeutic approaches. The review is clearly structured and written, its original and a research hot topic. Authors presented fundamental concepts to the reader to understand the deeply content that comes after the discussion. I have only minor concerns, as follows:
1) Figure 1 and 2 do not add some additional information. In general, Figures have no high resolution. This issue should be solved.
2) In table 2, some references are missing.
3) Please, remove empty line in Table 1.
4) In line 221 the sentence “Additional role of EVs in the CNScould be involvement in the removal of surplus protein…” should be corrected.
5) I found some double spacing in some parts of the manuscript. Please verify that.
6) I suggest to add numbers in each section. Moreover, I suggest to add a conclusion section/ future perspective in this research area.
Author Response
Reviewer 1
Comments and Suggestions for Authors
In the present review, the authors provide an overview about the neurodegeneration process caused mainly by alterations in chaperones systems and EVs performance leading to neuropathies. Moreover, authors also outline the potential of EVs as diagnostic biomarkers, as well as in EVs-based therapeutic approaches. The review is clearly structured and written, its original and a research hot topic. Authors presented fundamental concepts to the reader to understand the deeply content that comes after the discussion. I have only minor concerns, as follows:
Authors’ Reply: We thank the Reviewer for paying considerable attention to our manuscript. Many comments and suggestions have allowed us to improve our work and have confirmed the importance of what we intend to carry out.
comment #1: Figures 1 and 2 do not add some additional information. In general, Figures have no high resolution. This issue should be solved.
Authors’ Reply: We thank the Reviewer for comments and suggestions about Figures. We agree that figures do not add any extra information, however, we think that the figures for our manuscript can be helpful to create an overview or summarize some ideas that are running in more detail in the text. Here with the figure 1, we tried to provide an instant overview of the protein folding process and the role of the chaperones in the different stages, and with the figure 2 we intended to give a brief overview of the neurodegeneration process in terms of initiation and progression during aging and to emphasize the role of chaperones and EVs. We provide high resolution figures as suggested by the Reviewer.
comment #2: In table 2, some references are missing.
Authors’ Reply: We thank the Reviewer. We insert the requested references in the table 2.
comment #3: Please, remove empty line in Table 1.
Authors’ Reply: We thank the Reviewer. We remove the empty line in the table 1.
comment #4: In line 221 the sentence “Additional role of EVs in the CNS could be involvement in the removal of surplus protein…” should be corrected.
Authors’ Reply: We corrected the line (line 517-519).
comment #5: I found some double spacing in some parts of the manuscript. Please verify that.
Authors’ Reply: We verified and corrected the double spaces in the manuscript.
comment #6: I suggest to add numbers in each section. Moreover, I suggest to add a conclusion section/ future perspective in this research area.
Authors’ Reply: We would like to thank the reviewer for these useful and appropriate suggestions. We added numbers to each section as advised. We wrote a conclusion according to the valuable suggestion of the reviewer.
Reviewer 2 Report
The authors review existing evidences on the importance of chaperones and extracellular vesicles in the pathophysiology of neurodegenerative diseases. Whilst the subject is timely and the authors present lots of data on this issue, the manuscript suffers from considerable drawbacks.
1. The main problem is associated with many awkward sentences which can be hardly understood. The authors are encouraged to run carefully through the entire text and rewrite what is clumsy. The examples to be corrected may be found in between the following lines: 271-275, 316, 380-383, 559-560, and many others.
2. The authors have ignored the fact that brain ischemia (due to several reasons) may contribute to the development of Alzheimer’s disease. Is there a link between cerebral ischemia, chaperones and extracellular vesicles? Can any therapeutic approaches result from consideration of this etiological factor?
3. I would suggest that the authors separate animal and clinical data which are now mixed with one another. Such a mixture is very hard to follow.
4. L. 271 - what does "(88). (89-92") mean as regards citations? A similar error may be found in l. 312, 366 and 583.
5. Lots of typos make the text difficult to go through.
Author Response
Reviewer 2
Comments and Suggestions for Authors
The authors review existing evidences on the importance of chaperones and extracellular vesicles in the pathophysiology of neurodegenerative diseases. Whilst the subject is timely and the authors present lots of data on this issue, the manuscript suffers from considerable drawbacks.
Authors’ Reply: We would like to thank the Reviewer for this very precise and insightful review and we really appreciate all valuable and supportive comments and suggestions.
comment #1: The main problem is associated with many awkward sentences which can be hardly understood. The authors are encouraged to run carefully through the entire text and rewrite what is clumsy. The examples to be corrected may be found in between the following lines: 271-275, 316, 380-383, 559-560, and many others.
Authors’ Reply: We thank the Reviewer for comments and suggestions. We made many corrections of the awkward sentences to improve the readability of the manuscript.
comment #2: The authors have ignored the fact that brain ischemia (due to several reasons) may contribute to the development of Alzheimer’s disease. Is there a link between cerebral ischemia, chaperones and extracellular vesicles? Can any therapeutic approaches result from consideration of this etiological factor?
Authors’ Reply: We thank the Reviewer for the valuable comment suggestion. We discussed the possible role and relation of the cerebral ischemia in the Alzheimer’s disease in the therapeutic section (10; line 1246-1256).
comment #3: I would suggest that the authors separate animal and clinical data which are now mixed with one another. Such a mixture is very hard to follow.
Authors’ Reply: We really appreciate the Reviewer’s suggestions, however, due to the lack of a huge number of clinical data, we consider that the results should be presented along with animal data.
comment #4: L. 271 - what does "(88). (89-92") mean as regards citations? A similar error may be found in l. 312, 366 and 583.
Authors’ Reply: We thank the Reviewer. We corrected the wrong format of inserted references.
comment #5: Lots of typos make the text difficult to go through
Authors’ Reply: We did appreciate for Reviewer’s very carefully reading of the manuscript and helping us to upgrade the text. We corrected all typos.

Reviewer 3 Report
This review is devoted to the role of extracellular vesicles and molecular chaperones in the development of age-related neurodegenerative disorders of the central nervous system.
The review contains sections devoted to shaperone system, extracellular vesicles, the role of extracellular vesicles in the transport through blood-brain barrier, the connection between aging, chaperone system and extracellular vesicles, Alzheimer’s disease, Parkinson disease, Huntington’s disease, prion disease, and therapeutic applications of extracellular vesicles and chaperones in CNS diseases
The authors mentioned research done to study the effects of yeast Hsp104 shaperone in mammalian model of Huntington disease.
The paper would benefit if the authors would add a little bit more information about anti-prion systems (and their potential therapeutic application) of other than human organisms (indeed, yeast were shown to possess several anti-prion systems). The similarity of principles of amyloid and prion formation between human and yeast provides a hope that these systems could potentially benefit humans as well one day. Some insight for the discussion could be found for example here: DOI 10.1021/acs.biochem.7b01285 ; DOI 10.1016/bs.adgen.2015.12.003 .
The list of abbreviations used would benefit the paper.
Line 143. “. The protein folding performed by chaperonins occur via…” This sentence should be corrected for a grammatical error.
Lines 135-140. The sentence from these lines should be rewritten for clarity.
Lines 149-152. Sentences from these lines should be corrected for grammatical errors.
Table 2. The sentences should be corrected for grammatical errors. “• Rab GTPases proteins such as Rab27A, Rab27b, Rab35involve in exosomes secretion. (58). (59)”
“ADP-ribosylation factor 6 (ARF6) induces the activation of the phospholipase D (PLD) cause phosphorylation of the extracellular signal-regulated kinase (ERK) on plasma membrane, resulted in activation of myosin light chain kinase (MLCK) involved in MVs secretion (60).“
Lines 230-231. The sentence should be checked for a grammatical error.
Line 234. The word “knowledge” seems to be uncountable.
Line 269. A typo in word “protein” should be corrected.
Line 272-273. The sentence should be rewritten for clarity.
Line 273. “Senescence process are accompanied…” This sentence should be checked for a grammatical error.
Lines 275-278. The sentence should be checked for grammatical errors.
Lines 295, 302-304, 314-316, 331, 337, 411, 559. The sentences there should be checked for grammatical errors and typos.
Lines 332-334, 611-614. Sentences should be rewritten for clarity.
The paper can be accepted after minor revision.
Author Response
Reviewer 3
Comments and Suggestions for Authors
This review is devoted to the role of extracellular vesicles and molecular chaperones in the development of age-related neurodegenerative disorders of the central nervous system.
The review contains sections devoted to chaperone system, extracellular vesicles, the role of extracellular vesicles in the transport through blood-brain barrier, the connection between aging, chaperone system and extracellular vesicles, Alzheimer’s disease, Parkinson disease, Huntington’s disease, prion disease, and therapeutic applications of extracellular vesicles and chaperones in CNS diseases
The authors mentioned research done to study the effects of yeast Hsp104 chaperone in mammalian model of Huntington disease.
Authors’ Reply: We thank the Reviewer for the very carefully reading the manuscript and revision.
comment #1: The paper would benefit if the authors would add a little bit more information about anti-prion systems (and their potential therapeutic application) of other than human organisms (indeed, yeast were shown to possess several anti-prion systems). The similarity of principles of amyloid and prion formation between human and yeast provides a hope that these systems could potentially benefit humans as well one day. Some insight for the discussion could be found for example here: DOI 10.1021/acs.biochem.7b01285; DOI 10.1016/bs.adgen.2015.12.003.
Authors’ Reply: We thank the Reviewer for suggestions references. We added more information about anti-prion systems as recommended (line 989-997; 1150-1156).
comment #2: The list of abbreviations used would benefit the paper.
Authors’ Reply: We would like to thank the Reviewer for this comment. We prepared the list of abbreviation.
comment #3: Line 143. “. The protein folding performed by chaperonins occur via…” This sentence should be corrected for a grammatical error.
Lines 135-140. The sentence from these lines should be rewritten for clarity.
Lines 149-152. Sentences from these lines should be corrected for grammatical errors.
Table 2. The sentences should be corrected for grammatical
errors. “• Rab GTPases proteins such as Rab27A, Rab27b,
Rab35involve in exosomes secretion. (58). (59)”
“ADP-ribosylation factor 6 (ARF6) induces the activation of the
phospholipase D (PLD) cause phosphorylation of the
extracellular signal-regulated kinase (ERK) on plasma
membrane, resulted in activation of myosin light chain kinase
(MLCK) involved in MVs secretion (60).“
Lines 230-231. The sentence should be checked for a
grammatical error.
Line 234. The word “knowledge” seems to be uncountable.
Line 269. A typo in word “protein” should be corrected.
Line 272-273. The sentence should be rewritten for clarity.
Line 273. “Senescence process are accompanied...” This
sentence should be checked for a grammatical error.
Lines 275-278. The sentence should be checked for grammatical
errors.
Lines 295, 302-304, 314-316, 331, 337, 411, 559. The sentences
there should be checked for grammatical errors and typos.
Lines 332-334, 611-614. Sentences should be rewritten for
clarity.
Authors’ Reply: We corrected sentences in the table 2 and in the lines:
- 157-159.
- 159-163.
- 526-527.
- 529-531.
- 600-604.
- 604-606.
- 633-635.
- 640-644.
- 654-658.
- 673-676.
- 681-685.
- 817-820.
- 1059-1060.
- 676-679.
- 1139-1143.